# Application of ERAS Protocol after VATS Surgery for Chronic Empyema in Immunocompromised Patients

**DOI:** 10.3390/healthcare10040635

**Published:** 2022-03-28

**Authors:** Beatrice Leonardi, Caterina Sagnelli, Alfonso Fiorelli, Francesco Leone, Rosa Mirra, Davide Gerardo Pica, Vincenzo Di Filippo, Francesca Capasso, Gaetana Messina, Giovanni Vicidomini, Antonello Sica, Mario Santini

**Affiliations:** 1Department of Thoracic Surgery, University of Campania “Luigi Vanvitelli”, 80131 Naples, Italy; beatriceleonardi01@gmail.com (B.L.); lion_athanatos@hotmail.it (F.L.); rosamirra92@yahoo.it (R.M.); davide_pica@hotmail.it (D.G.P.); vincenzodifilippo16@libero.it (V.D.F.); francescocapasso93@gmail.com (F.C.); adamessina@virgilio.it (G.M.); giovanni.vicidomini@unicampania.it (G.V.); mario.santini@unicampania.it (M.S.); 2Department of Mental Health and Public Medicine, University of Campania “Luigi Vanvitelli”, 80131 Naples, Italy; caterina.sagnelli@unicampania.it; 3Department of Precision Medicine, University of Campania “Luigi Vanvitelli”, 80131 Naples, Italy; antonello.sica@fastwebnet.it

**Keywords:** empyema, ERAS, VATS

## Abstract

Enhanced recovery after surgery protocols have shown improved clinical outcomes after lung resection surgery, but their application after empyema surgery is still limited. We retrospectively evaluated the outcomes of an adapted enhanced recovery after surgery (ERAS) protocol for immunocompromised patients who underwent video-assisted thoracoscopic surgery (VATS) surgery for chronic empyema between December 2013 and December 2021. The patients were divided into an ERAS group and a conventional treatment group. Peri-operative data were collected and compared between the two groups. The primary outcome was post-operative length of stay. Secondary outcomes were post-operative pain and post-operative complications (air leaks, atelectasis). A total of 86 patients, 45 in the ERAS group and 41 in the non-ERAS group, were considered. Chest tube duration (6.4 ± 2.3 vs. 13.6 ± 6.8 days) and post-operative length of stay (7.6 ± 1.6 vs. 16.9 ± 6.9 days) were significantly shorter in the ERAS group. The volume of chest drainage (103 ± 78 vs. 157 ± 89 mL/day) was significantly smaller in the ERAS group. There were no significant differences in operative time, blood loss, need for transfusion, tube reinsertion and median VAS score. The incidence of air leaks and atelectasis was significantly reduced in the ERAS group, as was the need for bronchoscopic aspiration. The application of an ERAS protocol after empyema VATS surgery for immunocompromised patients improved the surgical outcome, reducing the post-operative length of stay and rate of complications.

## 1. Introduction

Pleural empyema in immunocompromised patients is a threatening condition that often needs prolonged hospitalization and is associated with high morbidity and mortality [1]. Chronic empyema is characterized by fibrotic reaction and purulent inflammation resulting in the formation of multiple adherences and loculations in the pleural cavity [2]. Left untreated, this process leads to lung entrapment and loss of the physiological pulmonary expansion with consequent quality of life impairment. Video-assisted thoracoscopic surgery (VATS) is indicated in the fibrinopurulent stage (stage II) and organizing stage (stage III) for debridement, decortication, and lavage of the pleural cavity [3].

The progressive replacement of open decortication with minimally invasive techniques for advanced empyema treatment showed better functional outcomes, decreased rate of peri-operative complications and reduced morbidity [4].

The peri-operative management of immunocompromised patients with empyema is challenging, as we usually encounter debilitated patients with underlying conditions that hinder the healing process. To promote post-operative recovery in these patients, we questioned if guidelines for enhanced recovery after surgery (ERAS) [5], first described in colorectal surgery and subsequently applied in various surgical fields, could be adapted to be implemented in the current management of patients that undergone VATS treatment for empyema.

ERAS protocol after lung surgery [6] includes recommendations for pre-operative, intra-operative, and post-operative management identified through systematic review based on the best available evidence. Since the ERAS protocol’s first description, several studies have investigated the potential benefits of their application after lung cancer surgery, with encouraging results [7,8,9].

At present, ERAS protocol application after empyema surgery has been evaluated by a few studies with a focus on tuberculous empyema, but investigation in this field remains scarce.

Our investigation aimed at evaluating the effects of an ERAS protocol application in immunocompromised patients that have undergone VATS surgery for empyema.

## 2. Materials and Methods

### 2.1. Study Population and Design

We conducted a retrospective monocentric study on immunocompromised patients treated with VATS surgery for empyema between December 2013 and December 2021 at our hospital.

Inclusion criteria were (I) stage II or III empyema; (II) immunocompromised state; (III) VATS surgery for empyema; (IV) positioning of two drainage tubes.

Exclusion criteria were as follows: (I) conversion to thoracotomy (II); positioning of a single drainage tube to minimize the risk of bias due to a different level of post-operative pain intrinsic in the procedure.

ERAS principles adapted to empyema have been implemented in our institution in the last 5 years for patients subjected to VATS treatment. The non-ERAS control group (non-ERAS group) is composed of patients treated before the implementation of our ERAS protocol (ERAS group).

Empyema diagnosis was based on a clinical history of recent pneumonia, fever, cough or dyspnea, thoracic pain and suggestive chest X-ray or CT scan findings (unilateral fluid collection, pleural thickening, split pleura sign, loculations).

The immunocompromised state was assessed through medical records.

We evaluated as immunocompromised a patient with an immunodeficiency disorder, as follows:Malignant neoplasms or hematological disease (e.g, marginal lymphomas, follicular lymphoma, myelomas, chronic lymphocytic leukemias, myelodysplasias, etc.) that induce immune dysfunction by inducing a deficiency of immune effector cells or dysfunction of activities such as alteration in the synthesis of antibodies, in detail, defect or deficiency in the production of antibodies;Human immunodeficiency virus (HIV) infection and acquired immunodeficiency syndrome (AIDS), which cause progressive depletion of CD4 T cells;Patients with chronic long-term diseases that damage the immune system, inducing immunosuppression (e.g., HCV- or HBV-related hepatopathy, poliomyelitis, neurological disorders, transplantation);Patients with a history of repeated hospitalization for infections in the last 3 months before the diagnosis of empyema;Patients with iatrogenic immunodeficiency resulting from treatments with drugs that suppress or block the immune system (e.g., cancer treatment, treatment received after an organ or stem cell transplant, high-dose corticosteroids, or treatments for autoimmune diseases).

All patients had stage II or III empyema and were hospitalized for their clinic conditions. All patients underwent the intervention of triportal VATS debridement, pleural cavity lavage and decortication (Figure 1) with an anterior approach. An electrothermal bipolar vessel sealing system (LigaSure, Medtronic, Dublin, Ireland) was used for debridement and hemostasis when needed.

The procedure ended with the positioning of 2 24 Fr/28 Fr drainage tubes, 1 anterior through the camera incision and 1 through the posterior incision. A digital chest drainage system (Thopaz, Medela AG, Baar, Switzerland) with suction of −20/−25 cm H_2_O was routinely applied 24 h after surgery to prevent blood loss in the immediate post-operative period, considering that significant decortication was performed.

After the surgery, broad-spectrum antibiotics were administrated to all patients, followed by targeted therapy appropriate to the microbial agent identified through pleural fluid culture. If necessary, due to clinical conditions, bronchoscopic aspiration of secretions was performed. Chest drainages were removed in all patients when re-expansion of the lung was achieved in the absence of air leaks and when the amount of fluid drained was less than 250 mL in 24 h. Lung expansion was evaluated through chest X-ray, while pleural fluid discharge and air leaks were quantitatively evaluated through the digital drainage system. Post-operative pain was assessed in all patients with the Visual Analogue Scale (VAS score), a reproducible means to quantify pain level ranging from zero (absence of pain) to 10 (maximum level of pain).

The patients were considered ready for discharge after chest drainage removal when their pain was adequately managed by oral analgesics and when they were able to perform self-care. Each patient was re-examined at the outpatient clinic on the 7th day after discharge. At one month after discharge, we requested a HRCT scan of the thorax that the patient brought at the follow-up visit.

The primary outcome was the post-operative length of stay, while secondary outcomes were post-operative pain and post-operative complications (air leaks, atelectasis).

### 2.2. ERAS Protocol

Since there is no standardized ERAS protocol specific for empyema surgery, we developed one identifying the recommendations more appropriate to empyema management based on clinical practice and existing ERAS recommendations for different surgical specialties. A summary of the ERAS protocol is reported in comparison with the conventional recovery group in Table 1.

In the ERAS group, before admission, the patients were informed about the surgical procedure and the post-operative course phases. Respiratory function was evaluated through spirometry, and all patients were encouraged to perform respiratory function exercises through an incentive spirometer before the surgical procedure. Nutritional status was evaluated pre-operatively through the Malnutrition Universal Screening Tool (MUST), and if necessary, oral supplements were administrated.

Regional anesthesia with a nerve blockage was performed combined with general anesthesia, and opioid-based analgesia was avoided in favor of a combination of acetaminophen and non-steroidal anti-inflammatory drugs (NSAIDs). Early ambulation along with early-stage drinking and eating were promoted.

Post-operative respiratory function exercises, consisting of incentive spirometry assisted breathing exercises, pursed-lip breathing, deep breathing exercises and inspiratory muscle training, were encouraged, and their execution was supervised by nurses and physicians. Moreover, we suggested regularly performing respiratory function exercises through an incentive spirometer after discharge.

### 2.3. Data Collection

Anamnestic data, along with intra-operative and post-operative data, were collected. Intra-operative data included operative time and blood loss. Post-operative data included volume of chest tube drainage, days of chest tube duration, post-operative length of stay, albumin administration, post-operative laboratory data measured 3 days post-operatively (albumin, total protein, hemoglobin, white blood cells), complications (including air leaks, atelectasis, readmission rate), median VAS score, need for transfusion of packed red blood cells, need for further procedures (chest tube reinsertion, bronchoscopic aspiration), and 30-day mortality rate.

All procedures performed were in accordance with international guidelines, including the Helsinki Declaration of 1975, revised in 1983, the rules of the Italian laws of privacy, and the local Ethics Committees named: “Comitato Etico Universita’ Degli Studi Della Campania “Luigi Vanvitelli”—Azienda Ospedaliera Universitaria “Luigi Vanvitelli”—Azienda Ospedaliera Rilievo Nazionale “Ospedali Dei Colli””, Naples, Italy (protocol number 766/2018). Each patient signed an anonymous informed consent for the use of their data for anonymous clinical investigations and scientific publications. At the baseline visit, each patient signed informed consent for the surgical procedure.

### 2.4. Statistical Analysis

The summary statistics of patients’ characteristics were tabulated either as mean ± standard deviation (SD) for continuous variables or as the number of patients and percentages for categorical variables.

Student’s *t*-test and chi-squared test were used to compare different variables, as appropriate. MedCalc statistical software (Version 12.3) (MedCalc Software, Ostend, Belgium) was used. A *p*-value < 0.05 was considered statistically significant.

## 3. Results

### 3.1. Patients’ Characteristics

A total of 86 patients were included in our study. Of these, 45 patients followed an ERAS protocol (ERAS group), and 41 patients followed a conventional recovery program (non-ERAS group). The demographic and baseline characteristics of the study population, including age, gender, immunosuppression status, body mass index (BMI), symptoms, pre-operative laboratory data and comorbidities, are summarized in Table 2. The two groups were comparable in terms of demographic characteristics, symptoms, comorbidities, and empyema stage.

However, in the control group, the non-ERAS group, we found that white blood cell count was significantly lower than the ERAS group (*p* = 0.001). Analyzing this finding, we could not find a direct association behind the lower white blood cell count in the non-ERAS group, and we tribute that to selection bias due to our relatively small sample. The main etiologic agents were *Streptococcus* spp., *Pseudomonas aeruginosa*, and *Candida* spp.

### 3.2. Post-Operative Course Comparison of the ERAS Group and Non-ERAS Group

The comparison between groups in terms of operative and post-operative data is reported in Table 3.

There were no significant differences between the ERAS group and the non-ERAS group regarding operative time, blood loss and median VAS score measured at 24, 48 and 72 h post-operatively. The volume of chest drainage per day was significantly lower in the ERAS group compared with the non-ERAS group (*p* = 0.01).

Chest tube duration was significantly shorter in the ERAS group (*p* < 0.001); likewise, post-operative length of stay was significantly shorter in the ERAS group (*p* < 0.001). Regarding post-operative laboratory data, there were no significant differences in hemoglobin, white blood cell count and total protein level, whereas albumin level was significantly higher in the ERAS group compared with the non-ERAS group (*p* = 0.02).

Accordingly, albumin administrations were significantly lower in the ERAS group (*p* < 0.001). There was no need for transfusion of packed red blood cells for any of the patients.

Complications included air leaks, which were observed in 8 (9%) patients, and atelectasis, which was observed in 15 (17%) patients. Air leaks and atelectasis were significantly lower in the ERAS group compared with the non-ERAS group (*p* = 0.01 and *p* = 0.0009, respectively).

Regarding post-operative morbidity, the patients in the ERAS group required less bronchoscopic aspiration than the non-ERAS group (<0.001).

No patient needed chest-tube reinsertion or reintervention. There were no cases of major cardiac (including cardiac failure, cardiac arrest, pericarditis), pulmonary (including bronchopleural fistula, ARDS) or other complications in either group.

Readmission to the thoracic surgery department was not necessary for any patient. The 30-day mortality rate was 0 percent in both groups.

## 4. Discussion

The surgical treatment of chronic pleural empyema aims at the complete evacuation of infected material and the re-expansion of the lung, achieved through debridement of adherences, drainage of fibrous pus collections and pleural decortication. Several studies proved that VATS decortication and debridement is not only a feasible treatment in stage II or III empyema, but less invasive and less likely to cause complications compared with open decortication [10,11]. After surgery, if full lung re-expansion is not achieved and bronchial secretions are not effectively expelled through cough, bronchoscopic aspiration may be needed to remove retained secretions in an attempt to resolve atelectasis [12,13].

An immunocompromised state is a known risk factor for pleural empyema, with high morbidity and mortality [14]. Pleural empyema in immunocompromised patients is generally secondary to a pulmonary infection, but in some cases, can present as a complication of pneumonectomy, lobectomy, or esophageal perforation with pneumomediastinum [15,16].

Immunocompromised patients are usually in a generally weakened condition and are more prone to become physically inactive due to repeated hospitalizations, hence the need to identify strategies to promote an active recovery from the debilitating disease that is chronic empyema.

Only a few studies evaluated the application of an ERAS protocol after empyema surgery, specifically after tubercolous empyema; Xia and colleagues [17] retrospectively analyzed 92 patients who underwent tuberculous empyema surgical treatment, including 45 patients in the ERAS group and 47 patients in the control group. The patients that followed an ERAS protocol had a shorter chest tube duration and length of stay compared to the control group, and the volume of chest drainage was smaller. It should be noted that the operation method was VATS for patients in the ERAS group and thoracotomy for patients in the control group. No significant differences were observed regarding post-operative complications and causes of readmission.

Pulle and colleagues [18] conducted a retrospective analysis on 243 patients with tuberculous empyema that underwent VATS surgical treatment, including 77 patients in the ERAS group and 166 patients in the control group. For all patients, surgery was initiated in VATS, but in some cases, conversion to thoracotomy was needed. In the ERAS group, intra-operative blood loss, chest tube duration, length of stay and prolonged air leaks were significantly reduced compared to the control group. Time to return to normal activity was reduced in the ERAS group, and the rate of overall complications, including wound infection, was significantly lower. Common ERAS recommendations in these studies included pre-operative patient counselling, pre-operative and post-operative respiratory exercise, pre-operative nutritional evaluation, negative suction on chest tubes, early oral feeding, early ambulation.

The results of our study highlighted a rapid lung re-expansion in the patients that followed the ERAS protocol, as evidenced by the reduced volume of chest drainage and reduced air leaks incidence that favored early drainage removal. Stimulated by post-operative respiratory exercise and early mobilization, the expulsion of bronchial secretions through cough was effective in the ERAS group, requiring less bronchoscopic aspiration. The ERAS group was not associated with greater post-operative pain. We observed a significantly shorter post-operative stay in the ERAS group, which in immunocompromised patients is particularly important as prolonged hospitalization may favor the incidence of nosocomial infections [19]. We have not encountered major complications in the study population. Reinsertion of chest tube drainage or reintervention was not necessary in any case, testifying to the overall positive outcome of the surgery.

In Figure 2, we reported the progression of lung re-expansion on the CT scans of a patient in the ERAS group.

Analyzing the recommendations that we included in our ERAS protocol, we focused our attention on the main factors that we think have played a prominent role in our experience: respiratory exercise and early mobilization. While these factors have been previously investigated for their role in promoting functional recovery after lung resection surgery, evidence about their role after empyema VATS surgery is still limited.

Pre-habilitation programs after lung resection surgery generally consist of a multimodal intervention including aerobic and resistance exercises, respiratory training, nutritional counseling, and psychological assistance [20,21].

Post-operative respiratory function exercise after lung resection consists of different breathing exercises, including incentive spirometry-assisted exercise and deep breathing exercise, as well as inspiratory muscle training [22,23]. Their role in increasing functional outcomes is still debated, but pre-habilitation seems to reduce hospital length of stay and post-operative complications [24,25,26].

Respiratory exercise’s role in both the conservative and surgical management of empyema is crucial. Even so, there is scarce evidence on the matter in recent years. In general, respiratory exercise is fundamental after lung resection surgery, with patients that pre-operatively are used to their full potential respiratory function and post-operatively need to adapt to a new, often reduced, functionality. Its role has been proved by several studies and has been widely accepted and encouraged in hospital recovery programs [27].

In patients with empyema, pre-operative respiratory function is usually impaired, and after the mechanical liberation of the lung from the adherences achieved with surgery, the functionality tends to improve, with a process that seems to be the inverse of the recovery after lung resection. Respiratory exercise not only increases this natural tendency but is also fundamental to stimulate cough and to expectorate secretions.

Early ambulation within 24 h from surgery is desirable after lung surgery as it favors lung re-expansion and mobilization of bronchial secretions while preventing complications of prolonged immobility [7], but the psychological aspect of being encouraged to walk shortly after surgery should be valued as well. Early ambulation’s role is universally recognized after surgery for its positive effects on pain control, prevention of post-operative thrombosis and, in general, prevention of complications in nearly all surgical specialties [28,29,30]. The patient after surgery is often afraid of early mobilization, partly for fear of experiencing pain and, specifically in thoracic surgery, for the discomfort caused by the thoracic drains. The role of nurses, physiotherapists, and physicians is of utmost importance in encouraging and supporting patients in this phase.

The role of negative-pressure suction is still controversial, but in empyema management, it serves to favor visceral and parietal pleura apposition and to promote the sealing of air leaks. The use of portable digital thoracic suction systems solves the mobility problem that derives from wall aspiration with conventional chest drainages [31].

The recommendations included in the ERAS protocol applied by Xia et al. [17] and Pulle et al. [18] are very similar to the ones applied in our study and are, in general, coherent with the ERAS guidelines after lung resection [6]. There are a few differences; in fact, the appliance of negative pressure suction to drainages is not suggested by ERAS protocol after lung resection, while after empyema surgery, this seems to be helpful. There is also a different approach to pre-habilitation since patients with empyema need to be treated as soon as possible; therefore, pre-habilitation is intended to be an intervention limited in time and with a focus on the education of the patient.

There are some limitations to our study. The most relevant is the retrospective and non-randomized nature of the research. We could not include information regarding respiratory function comparisons between the pre-operative and post-operative period, but this may be interesting for future studies.

## 5. Conclusions

The implementation of an ERAS protocol after empyema VATS surgery for immunocompromised patients, contributing to a shorter post-operative length of stay and to a reduced complication rate, may be beneficial for an active recovery and return to normal activities.

## Figures and Tables

**Figure 1 healthcare-10-00635-f001:**
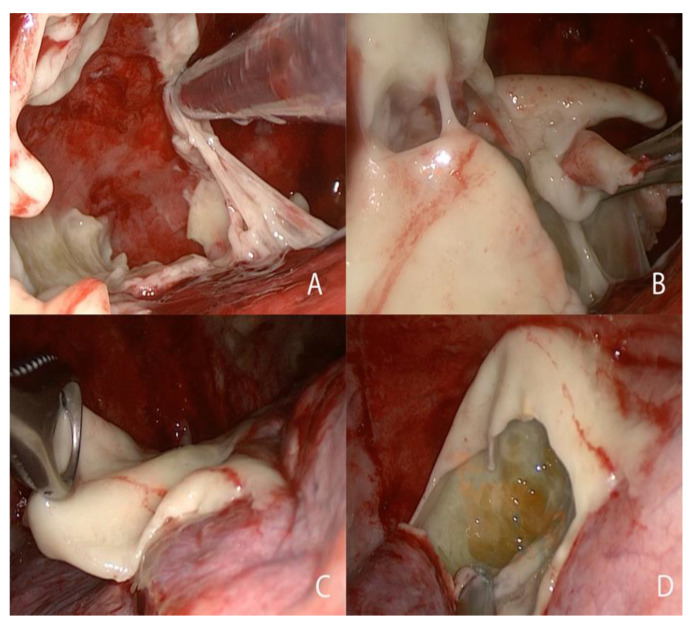
(**A**–**C**) VATS debridement and decortication. (**D**) Drainage of a pus cavity.

**Figure 2 healthcare-10-00635-f002:**
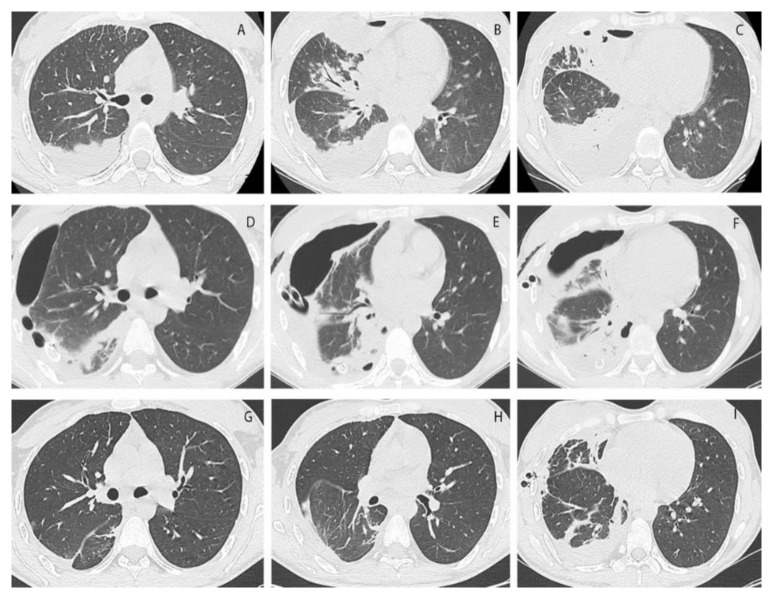
(**A**–**C**): Pre-operative CT scan; (**D**–**F**): first day post-operative CT scan, needed for the patient’s clinical conditions; (**G**–**I**): five days post-operative CT scan. In the last row, the drainages appear in a different position than the middle row because they were dislocated due to an abrupt movement of the patient. They were subsequently removed on that same day.

**Table 1 healthcare-10-00635-t001:** Comparison between the ERAS group and non-ERAS group.

Variables	ERAS Group	Non-ERAS Group
Pre-operative phase		
Preadmission information, education and counselling	Routine	Non-routine
Pre-operative respiratory function exercise	Routine	Non-routine
Fasting	Clear fluids allowed up until 2 h before anaesthesia, solids until 6 h before anaesthesia	All night fasting
Pre-operative nutritional evaluation	Routine	Non-routine
Intra-operative and post-operative phase		
Warming	Routine	Routine
Analgesia	Regional anaesthesia with intercostal nerve blockage, acetaminophen and NSAIDs combination. Opioid sparing analgesia	Acetaminophen and NSAIDs combination. Non-opioid sparing analgesia
Fluid management	Euvolemic fluid management	Non-routine
Early ambulation	Routine (12/24 h after surgery)	Non-routine
Early catheter removal	Routine	Non-routine
Early-stage drinking and eating	Routine	Non-routine
Post-operative respiratory function exercise	Routine, supervised	Non-routine
Negative pressure suction post-operatively	Routine	Routine

ERAS: enhanced recovery after surgery; NSAIDs: Non-steroidal anti-inflammatory drugs.

**Table 2 healthcare-10-00635-t002:** Demographics and pre-operative data.

Variables	Total (*n* = 86)	ERAS Group (*n* = 45)	Non-ERAS Group (*n* = 41)	*p*-Value
Age (mean ± SD)	52.6 ± 14.0	52.3 ± 13.0	53.0 ± 17.0	0.30
Gender (male), *n* (%)	71 (83%)	38 (84%)	33 (80%)	0.63
Marginal lymphomas, *n* (%)	15 (17.4%)	7 (15.5%)	8 (19.5%)	0.23
Follicular lymphoma, *n* (%)	12 (13.4%)	6 (13.3%)	6 (14.6%)	0.93
Myelomas, *n* (%)	15 (17.4%)	8 (17.8%)	7 (17.1%)	0.62
Chronic lymphocytic leukemias, *n* (%)	21 (24.4%)	11 (24.4%)	10 (24.3%)	0.84
Myelodysplasias, *n* (%)	7 (8.1%)	4 (8.9%)	3 (7.3%)	0.86
MGUS, *n* (%)	9 (10.4%)	4 (8.9%)	5 (11.9%)	0.88
Kidney transplantation, *n* (%)	5 (5.8%)	3 (6.7%)	2 (4.9%)	0.91
Repeated hospitalization for infections in the last 3 months before empyema, *n* (%)	2 (2.3%)	2 (4.4%)	0	/
Smokers (yes), *n* (%)	81 (94%)	43 (95%)	38 (93%)	0.57
BMI, Kg/m^2^ (mean ± SD):	23.2 ± 2.0	25.0 ± 3.7	21.1 ± 5.8	0.33
*Comorbidities, n (%):*				
Hypertension	15 (17%)	9 (20%)	6 (15%)	0.72
Diabetes	42 (49%)	24 (53%)	18 (43%)	0.09
COPD	16 (19%)	10 (22%)	6 (14%)	0.55
Cardiac	10 (12%)	7 (16%)	3 (7%)	0.23
*Pre-operative laboratory data (mean ± SD):*				
Albumin (g/dL)	3.5 ± 0.9	3.6 ± 0.7	3.4 ± 0.5	0.85
Total protein (g/dL)	6.9 ± 2.8	6.9 ± 2.1	6.3 ± 2.5	0.84
White blood cells (/uL)	16587 ± 1845	18360 ± 2352	13343 ± 1895	0.001
Hemoglobin (g/dL)	12.2 ± 2.8	12.4 ± 1.8	11.9 ± 3.9	0.41
*Symptoms, n (%):*				
Fever	74 (86%)	40 (89%)	34 (83%)	0.42
Cough	53 (62%)	29 (65%)	24 (58%)	0.57
Thoracic pain	76 (88%)	40 (89%)	36 (87%)	0.87
Dyspnea	85 (98%)	44 (97%)	41 (100%)	0.33
*Empyema stage II*	48 (56%)	25 (56%)	23 (56%)	0.96
*Empyema stage III*	38 (44%)	20 (44%)	18 (44%)	0.96

MGUS: monoclonal gammopathy of undetermined significance. COPD: chronic obstructive pulmonary disease. ERAS: enhanced recovery after surgery.

**Table 3 healthcare-10-00635-t003:** Operative and post-operative data.

Variables	Total (*n* = 86)	ERAS Group (*n* = 45)	Non-ERAS Group (*n* = 41)	*p*-Value
Operative time, min (mean ± SD)	95 ± 39	86 ± 44	106 ± 44	0.77
Blood loss, mL (mean ± SD)	261 ± 79	255 ± 54	285 ± 69	0.69
Chest drainage, mL daily (mean ± SD)	117 ± 39	103 ± 78	157 ± 89	0.01
Chest tube duration, days (mean ± SD)	9.7 ± 3.4	6.4 ± 2.3	13.6 ± 6.8	<0.001
Post-operative length of stay, days (mean ± SD)	11.8 ± 1.1	7.6 ± 1.6	16.9 ± 6.9	<0.001
Albumin administration, daily (mean ± SD)	1.3 ± 0.5	0.5 ± 0.1	1.8 ± 0.3	<0.001
Need for blood transfusion, *n* (%)	0	0	0	/
*Post-operative laboratory data* *(mean ± SD):*				
Albumin, g/dL	3.5 ± 0.9	3.9 ± 0.6	3.1 ± 0.7	0.02
Total protein, g/dL	6.9 ± 2.8	6.0 ± 2.1	5.5 ± 1.3	0.07
White blood, cells/uL	10,958 ± 1148	10,247 ± 1245	11,587 ± 1735	0.43
Hemoglobin, g/dL	10.3 ± 1.7	10.5 ± 2.8	10.1 ± 3.1	0.37
VAS score (mean ± SD)	2.9 ± 1.1	2.8 ± 0.9	3.1 ± 1.1	0.39
*Complications, n (%):*				
Air leaks	8 (9%)	1 (2%)	7 (17%)	0.01
Atelectasis	15 (17%)	2 (4%)	13 (31%)	0.0009
Readmission rate	0	0	0	/
*Further procedures, n (%):*				
Reinsertion of chest tube/reintervention	0	0	0	/
Bronchoscopic aspiration	0	1 (2%)	12 (29%)	<0.001
30-day mortality rate (%)	0	0	0	/

ERAS: enhanced recovery after surgery. VAS: Visual Analogue Scale.

## Data Availability

Not applicable.

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
