# Peer review of "Application of ERAS Protocol after VATS Surgery for Chronic Empyema in Immunocompromised Patients"

_healthcare, 2022, doi:10.3390/healthcare10040635_

Round 1

Reviewer 1 Report

In the paper by Leonardi et al. entitled"Application of ERAS protocol after VATS Surgery for Chronic Empyema in Immunocompromised patients" the authors evaluated the surgical outcome of ERAS protocol after empyema VATS surgery in immunocompromised patients. The article is well written and the results seem to point towards better management of this type of patients. I would just like to ask the authors if it was possible to have any information on the patients' immunosuppressive status and how this was determined. Why was the white cell count significantly lower in the control group than in the ERAS group? This aspect should be discussed.

Author Response

To the Editor in Chief of Healthcare
We re-submitted our article “Application of ERAS protocol after VATS surgery for chronic 
empyema in immunocompromised patients.”, Manuscript ID: -1582226, Section: Environmental Factors 
and Global Health, Special issue: Skin Disorders in Hematological Disease.
The following changes (shown underlined). The manuscript has been improved according to the suggestions 
of the reviewer:
Reviewer(s)' Comments to Author:
Reviewer 1: In the paper by Leonardi et al. entitled"Application of ERAS protocol after VATS Surgery for 
Chronic Empyema in Immunocompromised patients" the authors evaluated the surgical outcome of ERAS 
protocol after empyema VATS surgery in immunocompromised patients. The article is well written and the 
results seem to point towards better management of this type of patients.
Point 1: I would just like to ask the authors if it was possible to have any information on the patients' 
immunosuppressive status and how this was determined. 
Answer to the Reviewer point 1: The observation of the reviewer has been accepted and the table 2 and 
manuscript were modified accordingly. 
Point 2: Why was the white cell count significantly lower in the control group than in the ERAS group? This 
aspect should be discussed.
Answer to the Reviewer point 2: The observation of the reviewer has been accepted and the manuscript was 
modified accordingly.
Reviewer 2:
Point 1: What does it mean to be immunocompromised in the manuscript? Is there any type of scale or 
classification that people can use to determine their own relative risk for quantifying immunocompromised 
status?
Answer to the Reviewer Point 1: The observation of the reviewer has been accepted and the manuscript and 
the table 2 were modified accordingly.
Point 2: Although there is no standardized ERAS protocol specific for empyema surgery, there is Guidelines 
for enhanced recovery after lung surgery: recommendations of the ERAS and the ESTS. Is there any difference 
between your protocol and the recommendations?
Answer to the Reviewer Point 2: The observation of the reviewer has been accepted and the introduction 
section was modified accordingly.
Point 3: Is there any difference in the warming protocol between ERAS and control groups? Why do you not 
compare this between groups?
Answer to the Reviewer Point 3: The observation of the reviewer has been accepted and the manuscript was
modified accordingly. The warming protocol used was the same, using convective active warming devices, in 
both ERAS group and control group, so we did not compare this factor between groups. 
Point 4: With respect to the fast recovery, it is clear that the ERAS group is the statistical winner. However, 
there is no information about the empyema stage and severe infectious status before surgical intervention.
Answer to the Reviewer Point 4: The observation of the reviewer has been accepted and the table 2 and 
manuscript was modified accordingly. We also specified that all patients had clinic conditions that required 
hospitalization: “All patients had stage II or III empyema and were hospitalized for their clinic condition.” The 
infectious status was assessed in both groups through symptoms evaluation, laboratory data (table 2) in 
addition to evaluation of radiological findings, that led to empyema diagnosis and empyema stage evaluation.
Reviewer 3:
Summary: The manuscript Healthcare-1582226 describes a retrospective single institution study analyzing the 
outcome of immunocompromised adult patients undergoing video-assisted thoracoscopic decortication for 
empyema, using two different post-operative therapeutic protocols, initially the standard one followed by an 
Early Recovery After Surgery protocol (ERAS). The authors demonstrate that the ERAS protocol, which 
included preoperative teaching, preoperative lung expansion protocol, and nutritional assessment, as well as 
post-operative early ambulation, opioid sparing pain control approaches, and euvolemic fluid management, 
was associated with lower chest tube drainage fluid volume, shorter chest tube duration and shorter length of 
stay. Fewer air leaks and atelectasis episodes were also identified. The authors concluded that an ERAS 
protocol should be standard of care in immunocompromised patients undergoing VATS decortication for 
empyema.
Criticism:
Point 1: Study design: the study is a retrospective single institution cohort study of immunocompromised 
patients suffering from empyema. The study is well designed, and data collection include demographics and 
pre- and post-operative data. The authors do not include any specifics on the how they define the patients' 
immunocompromised state: the patients' white blood cell counts were elevated on the data shown. Can the 
authors provide any details on the diagnosis of immunocompromised state? Are the patients status post solid 
organ transplantation, bone marrow transplantation, undergoing chemotherapy for cancer, HIV+, on steroids 
or biological therapy for an auto-immune disease, etc? Are both groups matched for these diagnoses? Did the 
authors hold medications that worsened the immunocompromised state during the VATS procedure, if at all 
possible?
Can the authors describe the chest tube removal pathway they used in the study? Was it a standardized decisionmaking process, including the drainage volume, clinical assessment and radiological imaging, or was it 
physician dependent? The length of stay seems directly correlated to duration of chest tube, however in the 
control group, patients seem to have stayed in the hospital for an additional 3 days. How do the authors explain 
this? The authors did not include 30-day mortality and rate of readmission or need for further procedures, 
including re-insertion of chest tube in their data. The manuscript would be significantly improved if these data 
points are included. As a comment, early mobilization, chest physiotherapy and lung expansion protocols 
should be standard of care for any post-operative thoracic surgery patients, and this should have been 
implemented for the last 15-20 years.
The authors also comment quite extensively on pre-habilitation for thoracic surgery; It seems difficult to 
perform a pre-habilitation in this setting given that these patients diagnosed with empyema and being 
symptomatic (indication for VATS decortication) need a semi-urgent procedure.
Can the authors also include data on need for transfusion of packed red blood cells? Any differences between 
the groups? Blood loss was not different; however, their LOS was different, and albumin requirement.
Answer to the Reviewer Point 1: The observation of the reviewer has been accepted and the manuscript and 
table 2 and 3 were modified accordingly.
We specified the chest tube removal pathway, that was standardized for all patients, as well as the criteria used 
to discharge patients. We think that some patients in the control group, due to the increased rate of postoperative respiratory complications and the longer mean chest drainage permanence, required a longer clinical 
observation. We included data on 30-day mortality, rate of readmission and need for further procedures. 
We included data on the need for transfusion of packed red blood cells and discussed the albumin management.
Though albumin role in intensive care unit is controversial, we usually correct serum albumin levels in mild 
to severe hypoalbuminemia especially in patients with empyema, for its effect on maintaining the oncotic 
pressure since a great quantity of albumin is generally lost to the pleural fluid in this particular condition.
Point 2: Introduction: the introduction describes the background well. Further data on the inclusion criteria or 
immunocompromised state could be helpful.
Answer to the Reviewer Point 2: The observation of the reviewer has been accepted and the manuscript was 
modified accordingly.
Point 3: Patients and methods: This section is well described. Can the authors include the decision-making 
process for chest tube removal in the description and did this change between the control and the ERAS group? 
The internal ethics review study number should also be included.
Answer to the Reviewer Point 3: The observation of the reviewer has been accepted and the table 1 was 
modified accordingly.
Point 4: Results, tables and figures: results are well described. Further details on the diagnosis of 
immunocompromised state should be included. Thirty-day mortality and re-admission as well as additional 
procedure should be included. Further details on morbidity may also be helpful to better understand the 
differences between the two groups. It is not clear why these patients received albumin post-operatively, as 
this therapy seems controversial, at least in most centers in North America and has been shown not to change 
outcomes in general surgery patients in intensive care units. Can the authors comment on that?
Did the authors repeat multiple CTs on their patients? If yes, did this imaging change their process? In figure 
2, it seems that the CT scan image in "I" was not chronologically obtained at the same time as "G" and "H", as 
there seem to be still chest tube in place in "I" whereas there is no radiological evidence of chest tubes in "G" 
and "H". Please comment or correct.
Answer to the Reviewer Point 4: The observation of the reviewer has been accepted and the table e and 
manuscript were modified accordingly. We included data on 30-day mortality, rate of readmission and 
additional data on morbidities. 
We did not routinely perform multiple CT on our patients; usually our patients had a pre-operative CT scan of 
the thorax on which we assessed the diagnosis of empyema and planned surgery, and a control CT scan one 
month after discharge. We requested additional CT scans during the hospital stay only for clinical reasons, in 
the specific case of the patient in the Figure 2 we needed to evaluate if a bronchoscopy was needed due to the 
patient clinical conditions at the moment. Images “G”, “H” and “I” are obtained from the same CT scan 
performed on fifth postoperative day, after the drainages were accidentally partially pulled out by the patient 
during an abrupt movement. For this reason, on the CT scan they appear in a different position and are visible 
only in “I” image. They were subsequently removed on that same day.
Point 5: Discussion: The discussion is well written and explains the current findings and the literature well. 
The authors could discuss the impact of co-morbidities and complications further. Early ambulation and lung 
expansion protocols have been used for more than a decade and the authors may comment on that further. 
Given the many different causes of immunocompromised state, they should discuss the originality of their 
findings.
Answer to the Reviewer Point 5: The observation of the reviewer has been accepted and the manuscript was 
was modified accordingly.
Point 6: Minor comment: A few grammatical and typing errors should be corrected.
Answer to the Reviewer Point 6: The observation of the reviewer has been accepted and the manuscript has 
been evaluated by an expert of English language.
We thank the Editor and the Reviewers for helping us to improve our paper. 
The manuscript has been read and approved by all the authors. 
We also declare that we have no conflict of interest in connection with this paper.
We sincerely hope that the enclosed manuscript can be accepted for publication in the: Healthcare
Prof. Alfonso Fiorelli

Reviewer 2 Report

1.What does it mean to be immunocompromised in the manuscript? Is there any type of scale or classification that people can use to determine their own relative risk for quantifying immunocompromised status?

2.Although there is no standardized ERAS protocol specific for empyema surgery, there is Guidelines for enhanced recovery after lung surgery: recommendations of the ERAS and the ESTS. Is there any difference between your protocol and the recommendations?

3.Is there any difference in the warming protocol between ERAS and control groups? Why do you not compare this between groups?

4.With respect to the fast recovery, it is clear that the ERAS group is the statistical winner. However, there is no information about the empyema stage and severe infectious status before surgical intervention.

Author Response

(The authors gave the same response as above.)

Reviewer 3 Report

Summary: 

The manuscript Healthcare-1582226 describes a retrospective single institution study analyzing the outcome of immunocompromised adult patients undergoing video-assisted thoracoscopic decortication for empyema, using two different post-operative therapeutic protocols, initially the standard one followed by an Early Recovery After Surgery protocol (ERAS). The authors demonstrate that the ERAS protocol, which included preoperative teaching, preoperative lung expansion protocol, and nutritional assessment, as well as post-operative early ambulation, opioid sparing pain control approaches, and euvolemic fluid management, was associated with lower chest tube drainage fluid volume, shorter chest tube duration and shorter length of stay. Fewer air leaks and atelectasis episodes were also identified. The authors concluded that an ERAS protocol should be standard of care in immunocompromised patients undergoing VATS decortication for empyema. 

Criticism:

  1. Study design: the study is a retrospective single institution cohort study of immunocompromised patients suffering from empyema. The study is well designed, and data collection include demographics and pre- and post-operative data. The authors do not include any specifics on the how they define the patients' immunocompromised state: the patients' white blood cell counts were elevated on the data shown. Can the authors provide any details on the diagnosis of immunocompromised state? Are the patients status post solid organ transplantation, bone marrow transplantation, undergoing chemotherapy for cancer, HIV+, on steroids or biological therapy for an auto-immune disease, etc? Are both groups matched for these diagnoses? Did the authors hold medications that worsened the immunocompromised state during the VATS procedure, if at all possible?
    Can the authors describe the chest tube removal pathway they used in the study? Was it a standardized decision-making process, including the drainage volume, clinical assessment and radiological imaging, or was it physician dependent? The length of stay seems directly correlated to duration of chest tube, however in the control group, patients seem to have stayed in the hospital for an additional 3 days. How do the authors explain this? The authors did not include 30-day mortality and rate of readmission or need for further procedures, including re-insertion of chest tube in their data. The manuscript would be significantly improved if these data points are included. As a comment, early mobilization, chest physiotherapy and lung expansion protocols should be standard of care for any post-operative thoracic surgery patients, and this should have been implemented for the last 15-20 years. 
    The authors also comment quite extensively on pre-habilitation for thoracic surgery; It seems difficult to perform a pre-habilitation in this setting given that these patients diagnosed with empyema and being symptomatic (indication for VATS decortication) need a semi-urgent procedure.
    Can the authors also include data on need for transfusion of packed red blood cells? Any differences between the groups? Blood loss was not different; however, their LOS was different, and albumin requirement.
  2. Introduction: the introduction describes the background well. Further data on the inclusion criteria or immunocompromised state could be helpful. 
  3. Patients and methods: This section is well described. Can the authors include the decision-making process for chest tube removal in the description and did this change between the control and the ERAS group? The internal ethics review study number should also be included. 
  4. Results, tables and figures: results are well described. Further details on the diagnosis of immunocompromised state should be included. Thirty-day mortality and re-admission as well as additional procedure should be included. Further details on morbidity may also be helpful to better understand the differences between the two groups. It is not clear why these patients received albumin post-operatively, as this therapy seems controversial, at least in most centers in North America and has been shown not to change outcomes in general surgery patients in intensive care units. Can the authors comment on that? 
    Did the authors repeat multiple CTs on their patients? If yes, did this imaging change their process? In figure 2, it seems that the CT scan image in "I" was not chronologically obtained at the same time as "G" and "H", as there seem to be still chest tube in place in "I" whereas there is no radiological evidence of chest tubes in "G" and "H". Please comment or correct. 
  5. Discussion: The discussion is well written and explains the current findings and the literature well. The authors could discuss the impact of co-morbidities and complications further. Early ambulation and lung expansion protocols have been used for more than a decade and the authors may comment on that further. Given the many different causes of immunocompromised state, they should discuss the originality of their findings. 
  6. Minor comment: A few grammatical and typing errors should be corrected.

Author Response

(The authors gave the same response as above.)

Round 2

Reviewer 2 Report

The authors have well replied to comments from the first round of reviews.

Author Response

To the Editor in Chief of Healthcare

We re-submitted our article “Application of ERAS protocol after VATS surgery for chronic empyema in immunocompromised patients.”, Manuscript ID: -1582226, Section: Environmental Factors and Global Health, Special issue: Skin Disorders in Hematological Disease.

The following changes (shown underlined). The manuscript has been improved according to the suggestions of the reviewer:

Reviewer(s)' Comments to Author:

Reviewer 2:

Point 1: English language and style are fine/minor spell check required

Answer to the Reviewer Point 1: The observation of the reviewer has been accepted and the manuscript has been evaluated by an expert of English language.

Reviewer 3:

Point 1:  On figure 2, the figure-legend should explain why there are still chest tubes in the subcutaneous space.

Answer to the Reviewer Point 1: The observation of the reviewer has been accepted and the manuscript was modified accordingly.

Point 2: A few spelling and grammatical errors should be corrected.

Answer to the Reviewer Point 2: The observation of the reviewer has been accepted and the manuscript has been evaluated by an expert of English language.

Point 3: The paragraph describing the immunocompromised state of patients (Lines 87-93) needs to be rewritten and clarified.

Answer to the Reviewer Point 3: The observation of the reviewer has been accepted and the manuscript was modified accordingly.

Point 4: The VAS score is not defined nor is its acronym, that is used once in the abstract and 3 times in the paper. Please define it for the readership and briefly explain its use (i.e. reproducible and quantifiable pain assessment and score).

Answer to the Reviewer Point 4: The observation of the reviewer has been accepted and the manuscript was modified accordingly.

Point 2: In conclusion, the manuscript has been significantly improved and some minor revisions are suggested. The feasibility and benefit of an ERAS protocol in this highly selected population is clearly demonstrated.

Answer to the Reviewer Point 2: The observation of the reviewer has been accepted and the manuscript was modified accordingly.

We thank the Editor and the Reviewers for helping us to improve our paper.

The manuscript has been read and approved by all the authors.

We also declare that we have no conflict of interest in connection with this paper.

We sincerely hope that the enclosed manuscript can be accepted for publication in the: Healthcare

Prof. Alfonso Fiorelli

Reviewer 3 Report

The manuscript Healthcare-1582226 describes a retrospective single institution study analyzing the outcome of immunocompromised adult patients undergoing video-assisted thoracoscopic decortication for empyema, using two different post-operative therapeutic protocols, initially the standard one followed by an Early Recovery After Surgery protocol (ERAS). The authors demonstrate that the ERAS protocol, which included preoperative teaching, preoperative lung expansion protocol, and nutritional assessment, as well as post-operative early ambulation, opioid sparing pain control approaches, and euvolemic fluid management, was associated with lower chest tube drainage fluid volume, shorter chest tube duration and shorter length of stay. Fewer air leaks and atelectasis episodes were also identified. The authors concluded that an ERAS protocol should be standard of care in immunocompromised patients undergoing VATS decortication for empyema. 

Criticism:

The study has been significantly improved and most questions were answered appropriately. The inclusion criteria and patient immunocompromised details are helpful. 

On figure 2, the figure-legend should explain why there are still chest tubes in the subcutaneous space. 

A few spelling and grammatical errors should be corrected. 

The paragraph describing the immunocompromised state of patients (Lines 87-93) needs to be rewritten and clarified. 

The VAS score is not defined nor is its acronym, that is used once in the abstract and 3 times in the paper. Please define it for the readership and briefly explain its use (i.e. reproducible and quantifiable pain assessment and score). 

In conclusion, the manuscript has been significantly improved and some minor revisions are suggested. The feasibility and benefit of an ERAS protocol in this highly selected population is clearly demonstrated. 

Author Response

(The authors gave the same response as above.)
